# The Path of 'No' Resistance to Temptation: Lessons Learned from Active Buddhist Consumers in Thailand

**Apiradee Wongkitrungrueng [1] and Panitharn Juntongjin [2,\*]**

[1] Business Administration Division, Mahidol University International College, Nakhonpathom 73170, Thailand
[2] Department of Marketing, Chulalongkorn Business School, Chulalongkorn University, Bangkok 10330, Thailand
[\*] Correspondence: panitharn@cbs.chula.ac.th

**Abstract:** Mindfulness practice and mindful consumption have increasingly attracted the interests of academics and the general public worldwide. Despite the fact that mindfulness meditation has its roots in Buddhism, little empirical research has studied mindfulness and mindful consumption from the Buddhist principles and from the perspective of active Buddhists who regularly dedicate themselves to Buddhist practice with the goal of achieving liberation from suffering. This study builds on and extends previous research that established a research agenda regarding how mindfulness could transform consumer behavior and lead to higher levels of well-being. The purpose of this study is to gain an understanding of the ways in which active Buddhist consumers living in the city have disengaged from the consumerist culture and altered their lifestyle and consumption behaviors. To grasp the subtle complexity of the experience, fifteen active Buddhist practitioners were interviewed in depth. As a result of this, a Buddhist-based behavioral change model with seven stages is developed. Drawing on behavioral change models, such as the transtheoretical model (TTM) and the stepwise model of behavior change (SMBC), this model aims to demonstrate how active Buddhist consumers have transformed their consumption behavior patterns and overcome temptation without resistance. The transformative mechanism and consumer strategies were also extracted to provide lessons learned and management implications.

**Keywords:** Buddhism; religious practices; self-control; mindfulness; consumerism; mindful consumption

## 1. Introduction

Despite the consumer sustainability trend, in which consumers seek to live more sustainably, and the COVID-19 pandemic that significantly changed consumer behavior patterns to be more mindful in their decision-making and focus more on locally produced, greener, and functional products (KPMG International 2020), consumer spending is estimated to grow almost 50% higher in 2030 than in 2020 (Fengler and Kharas 2021). The culture of consumerism is accelerated by technological advances that have made it more convenient for consumers to shop at any time from anywhere in the world. However, the increase in consumption as a means of (temporary) life fulfillment, stimulated by advertisers, has come at the expense of the well-being of individuals, society, and the planet (Rosenberg 2004). Policymakers, NGOs, and academics have faced challenges in changing consumer behavior (Prothero et al. 2011) and a shift away from the consumerist culture seems impossible (Rosenberg 2004).

Lennerfors (2015) proposed Buddhist economics as a desirable alternative to market capitalism and consumerism, since Buddhist principles are found to tame materialism (Belk 2011; Kieschnick 2003; Pace 2013) and inspire the development of an economically and ecologically sustainable future (Schumacher 1973; Sivaraksa 2009). Unlike modern economics, which seeks to maximize consumption through the optimal pattern of productive effort, Buddhist economics aims to maximize human well-being through an optimal method of

consumption that causes no harm to oneself or others (Schumacher 1973; Payutto 1994). Buddhist economics requires ethical awareness and spiritual response to emergent desires in daily life (Mutakalin 2014). Thus, vital to its implementation is mindfulness practice, cultivated within the Buddhist principle, e.g., the Eightfold path (Brown 2015; Wintrobe 2019).

A growing body of literature on mindfulness in the consumption domain (e.g., Bahl et al. 2016; Fischer et al. 2017; Rosenberg 2004; Sheth et al. 2011) suggests the pivotal role of mindfulness in transforming consumer behavior and empowering them to make optimal choices. However, prior research remains largely conceptual and descriptive (Gupta et al. 2021). Little empirical research examines mindfulness's role in consumption (Bahl et al. 2016; Mick 2017; Milne et al. 2020). There are specific questions about how mindfulness practice influences consumption behaviors and contributes to individual behavior change (Thiermann and Sheate 2021), how it aids in temptation control (McCullough and Willoughby 2009), and the underlying mechanism by which mindfulness enables consumers to make more deliberate choices (Bahl et al. 2016). The extant empirical research is criticized for using quantitative measurement and mindfulness interventions in a short time frame with convenience samples and biased populations rather than general population samples with formal meditation experience (Thiermann and Sheate 2021). A qualitative method is needed to gain a nuanced understanding of the connection between mindfulness and consumption (Thiermann and Sheate 2021).

The current study responded to the need for empirical and qualitative research with Buddhist consumers who regularly and actively follow Buddhist principles, which are not limited to mindfulness meditation. Feng et al. (2018) noted the difference between mindfulness in Western psychology and Buddhist mindfulness, the latter of which is relatively underexamined. This paper builds on and contributes to previous research (e.g., Bahl et al. 2016; Mick 2017; Thiermann and Sheate 2021) that established a research agenda for ways that mindfulness could transform consumer behavior and lead to higher levels of well-being. This paper delineates the role of Buddhist practice in enabling active Buddhist practitioners to manage their temptations and transform their consumption behavior patterns. Drawing on existing behavior change models, such as the transtheoretical model (TTM), the paper develops a Buddhist-based interpretation of the behavior change model to understand the process and uncover the underlying mechanism of transformative consumption behaviors. Our findings provide additional insights into prior research (e.g., Bahl et al. 2016; Eckhardt 2011; Pongsakornrungsilp and Pusaksrikit 2011; Pace 2013; Rosenberg 2004; Thiermann and Sheate 2021) by revealing motivation and consumption patterns of active Buddhist practitioners who, unlike lay Buddhists, wish to liberate themselves from suffering and experience significant changes in their consumption motives and behaviors following consistent Buddhist practices. We seek to understand how they let go of their desires for products and attachment to pleasure that the products provide, as well as how they adapt their lives and consumption habits, in order to identify the underlying mechanism of their behavior transformation.

To understand the pivotal role of Buddhist practice in mitigating mindless consumption, we first provide the background of consumerism and connect it with the Buddhist concept (Paṭiccasamuppāda), describing the cause of suffering, which is the foundational tenet of Buddhism. We then review self-control strategies and mindfulness-related research to identify research gaps and establish research questions that will be addressed in this study.

## 2. Conceptual Background

### 2.1. Consumerism and the Vicious Cycle

Consumerism is a modern way of life, prevailing in the last few decades of the twentieth century, when basic material needs are generally met (Mutakalin 2014). Consumerism is associated with the use of strategies and techniques that encourage consumers to consume more to expand their needs and desires (Packard 1957). The overemphasis on customers' needs, wants, and desires have laid the foundation for another definition of consumerism,

"consumption as a means for happiness and well-being" (Yani-de-Soriano and Slater 2009). Thus, consumerism is an ideology that encourages the acquisition and consumption of goods and services (McIlhenny 1990; Swagler 1997). Society believes that consumption positively impacts economic growth and individual welfare; however, it is accompanied by adverse effects, such as private and public debt and the depletion of natural resources (Goodwin et al. 2008). In a consumer society, when consumers decide to buy a product, they consider not only what it can do but also what it means, how it makes them feel (Levy 1959), and how it satisfies the desire for knowledge (Berlyne 1970). Products, therefore, provide functional, social, emotional, and epistemic value (Sheth et al. 1991). Through products, consumers can enjoy sensorial and emotive experiences (hedonic consumption), satisfy curiosity and cognitive needs, and acquire and display status (compensatory and conspicuous consumption) (Hoyer and MacInnis 2008). In a consumer-led approach to consumerism, consumption is a broader process of social communication and identity construction (Mutakalin 2014).

Rosenberg (2004) suggested two main problems in consumerism. The first problem stems from the fact that humans tend to process information automatically or unconsciously (i.e., system 1), rather than slowly and deliberately (system 2) (Kahneman 2011). Therefore, most consumer behaviors are automatic and mindless. Marketers capitalize on this automaticity through urgency and scarcity tactics, mere exposure, and conditioning (associating the product with desirable outcomes) to shape consumer preference. The second problem is related to the motivation behind consumption: consumers consume goods to quickly satisfy the need for fulfillment. Over the past century, capitalism, a market economy, and consumerism seem to contribute to a false sense of individualism and inner emptiness (Rosenberg 2004). Marketers seek to reassure or soothe consumers with products and offer the fantasy that the consumer's life can be transformed into a glorious, problem-free life, like the model in the advertisement (Cushman 1990). Consumerism suggests that consumers consume to satisfy new needs created continuously by marketers to pursue happiness and products become sham objects or signs of happiness (Baudrillard 1998). Yet, products do not have the real power to bestow happiness and well-being (Diener 2009). Consumers always desire more and more products because their aspirations are heightened (i.e., hedonic adaptation) (Lyubomirsky 2011; Brickman and Campbell 1971), resulting in a vicious cycle. Therefore, the desire for happiness underlying consumption is not satisfied by purchasing products (Malpas 2005). However, consumers may not be aware of such a fact and keep buying products to meet specific needs and values. Such ignorance leads to delusion and suffering, according to *Paṭiccasamuppāda* (i.e., the chain of dependent origination), the Buddha's most complete analysis of the conditions leading to suffering and suffering's end (Thanissaro 2008).

Paṭiccasamuppāda is a natural system of the way delusion and suffering arise in human beings (Watts and Loy 2002; Mahāsi Sayādaw 2017) (see Figure 1). The process starts with the nature of consumers that tend to ignore three fundamental characteristics of existence and beings (i.e., suffering (*duḥkha*), impermanence (*c*), and no-self (*anattā*)). In other words, human life is unsatisfactory, involving physical and mental discomfort because everything (e.g., objects, self, and thoughts/feelings) is impermanent and ever-changing (Mick 2017; Cayton 2012). The cause of suffering is ignorance of the sources of suffering, which is craving/desire (*taṇhā*) for physical sensations, the continued existence of pleasant things, and the non-existence of unpleasant things in life (*taṇhā*) (Dalai Lama 2005; De Silva 1990; Mick 2017). With ignorance, craving blinds them to suffering and creates the illusion of happiness (Mahāsi Sayādaw 2017). Conditioned with this ignorance, misconception, or illusion (*avijjā*), consumers believe shopping and consumption are pleasing. Furthermore, they tend to fabricate their thoughts and feelings (mental formation) (Thanissaro 1997), such as not seeing a product as it is but instead fabricating a product as a symbol or a means toward defining and fulfilling one's self and, thus, possessing the product to fulfill their desires and selves.

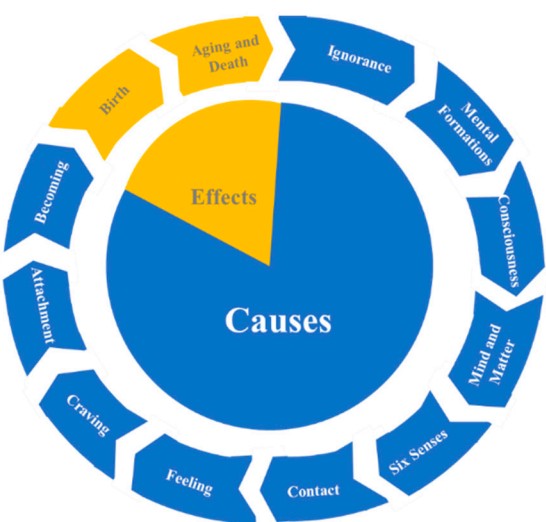

**Figure 1.** Paṭiccasamuppāda: A Discourse on Dependent Origination (adapted from Sayādaw 2017).

With such mental formation/fabrication (*saṅkhāra*), consumers become conscious of (*vijñāna*) products that can bring pleasure and value to them. Thus, when our sensory systems (eyes, ears, nose, tongue, body, and mind) (*saḷāyatana*) come into contact (*sparśa*) with consumable material objects (*nāmarūpa*), they create sensory experiences (sight, sound, smell, taste, touch, and mental experiences). When consumers receive those sensory experiences, they assign the meaning based on their memory/experience (*saññā*) and, consequently, feelings (*vedanā*) (i.e., pleasant, painful, neutral) of those experiences arise. When consumers are exposed to an advertisement or a product, think it is valuable, and feel positive about it, they then desire (*taṇhā*) it to please their senses, enhance their identity, or help avoid unpleasant feelings. The craving for consumable objects or states of being becomes stronger when consumers compare with their future or past self and with others and want things to be different and better than they are (i.e., the comparing and judging consumer mind, termed by Mick 2017). The difference from such a comparison creates the experience of lack and fear consumers feel inside, thus, resulting in greater craving to maintain the desired self. Cravings lead to sensual, attitudinal, and behavioral clinging/attaching (*upādāna*) to material forms, mental states, and the self. Such an attaching and depending consumer mind (Mick 2017) further intensifies ego and delusive identity (*bhāva* and jati). When our identity is challenged due to aging and death (*jarāmaraṇa*) of our possessions and extended selves (Belk 1988), consumers experience various forms of physical, emotional, interpersonal, and economic suffering (*duḥkha*) (e.g., pain, despair, grief, sorrow, or lamentation) (Payutto 1995). Modern humans are obsessed with their possessions (Preston and Vickers 2014), such that losing a possession is a loss of self (Fromm 1976). There are also numerous forms of excessive and addicted behaviors (Mick 2017), reflecting the pervasiveness of attachments and dependencies, which ultimately affect consumers' well-being. Paṭiccasamuppāda explains how consumers become attached to consumption, which can cause suffering. To break such a continuous chain of suffering, the links between the chain's elements have to be severed to end the vicious cycle of feelings developing into desires or cravings, which eventually lead to attachment or clinging.

### 2.2. Self-Control Strategies Defending against Consumerism

To resist temptation and break the vicious cycle, psychological and consumer research suggest that consumers must exert self-control, a struggle between compelling forces (e.g., impulses and desires) and restraining forces (e.g., self-regulatory goals). Consuming goods and services for other reasons beyond meeting physical needs can make people feel good, but feeling good often involves a feeling of guilt (Lascu 1991). Thus, consumers attempt to justify the chosen option (Dhar and Wertenbroch 2000) or to control hedonic temptations (Wertenbroch 1998). On the one hand, consumers may justify an indulgent

purchase by adopting tactics that reduce the guilt or negative attributions or facilitate the purchase. These tactics include reward justification (O'curry and Strahilevitz 2001), effortful consumption requirement (Kivetz and Simonson 2002), and committing to a virtuous act before indulging (Khan and Dhar 2006). On the other hand, to control temptations, consumers must arduously construct counterarguments regarding such abstract concepts as the affordability and practicality of the item and the future impact of purchasing it (Malter 1996) and the necessity of the purchase (Shehryar et al. 2001). Other strategies typically include temptation avoidance (Hofmann et al. 2010), powerful distractors (Florsheim et al. 2008), and effortful inhibition of temptation, such as suppressing thoughts or forcing oneself to concentrate (Baumeister 2002), lowering the priority of a specific product on the shopping list, berating the available choice or enhancing the value of a present possession, and postponing the purchase based on more information searching and feedback from others or future deal, product choice, and technology (Shehryar et al. 2001).

Resisting temptations depends on the *individual's goals, goal monitoring, and capacity for self-control*; if one component fails, self-control will fail. The first component can fail when consumers are emotionally upset. The goal of feeling better through consuming goods or services would precede other self-regulatory goals. When consumers fail to keep track of their behaviors towards their self-regulatory goals, they are more likely to fall prey to desires. Self-control depends very much on the individual's strength or willpower, which can be depleted and replenished in the short run; therefore, willful suppression of desire tends to backfire (Johnston et al. 1999). Yet, regular mindfulness practice can further develop willpower (Muraven et al. 1999), empowering people to disengage from automatic thoughts, desires, habits, and unhealthy behavior patterns by accepting them as transient events that will eventually fade (Brown and Ryan 2003; Hofmann and Van Dillen 2012). McCullough and Willoughby (2009) proposed that religion can influence self-regulation by influencing goals, promoting self-monitoring, and building self-regulatory strength. Rounding et al. (2012) found that implicit reminders of religious concepts refueled individuals' capacity to exercise self-control. However, much is still unknown about how religious concepts and practices improve one's self-regulation.

### 2.3. Mindfulness and Buddhist Consumption

Derived from Buddhist origins, mindfulness refers to deliberate, unbiased, open-hearted awareness of perceptible experience in one's mind (including emotions and sensations provided from one's body and the outside world) in the present moment (Fischer et al. 2017; Rosenberg 2004). To practice mindfulness, the Buddha suggested contemplating the four objective domains that comprise the entire field of human experiences (body, feelings, states of mind, and Dhammas) and seeing them as they are, the body/feelings/mind as the body/feelings/mind, not my body/feelings/mind, and the Dhamma/Buddha's teaching is within us (Gunaratana 2012). Table 1 summarizes how the four foundations unfold in a definite sequence, starting with the body as the coarsest until reaching the last, concerned with the exploration of practice experience reflecting the Buddha's teaching, which is subtlest (Bodhi 2021).

**Table 1.** Foundations of Buddhist Mindfulness (*Satipaṭṭhāna Sutta*).

| Object | Description of Mindfulness Practice |
|---|---|
| 1. Body | Nonjudgmental awareness, attention, or focus of <br><br>• breathing (in-out, short-long breathing) <br>• postures (e.g., walking, standing, sitting, lying down) and action (looking up-down, reading, bending, eating, falling asleep) so one knows what one's doing and why one's doing it <br>• body's real nature (thirty-two body parts comprise various organs, tissue, and bodily fluid) is unattractive, and the body's primary elements (earth, water, air, fire) show that a body is nothing more than a set of material processes |
| 2. Feelings | • feeling (e.g., pleasant, painful, or neutral) that arises and dissolves, showing that feeling is impermanent |
| 3. States of Mind | • states of mind (e.g., liberated, bound, with/without greed, aversion, delusion) shaped by external factors affecting sensory systems and internal factors (e.g., memory, imagination, daydreaming) show the runaway train of unsatisfactory thoughts and help discern the difference of feelings when the mind is/is not attached. |
| 4. Dhamma | • five hindrances to mindfulness practice (e.g., drowsiness and dullness, ill will, restlessness and worry, doubt, sensual desire) <br>• Antidotes to five hindrances (e.g., mindfulness, investigation of phenomena, diligence, joy, tranquility, concentration, and equanimity |

Source: Bodhi (2021) and Gunaratana (2012).

How mindfulness could promote changes in consumption behaviors has been discussed conceptually by Hölzel et al. (2011), Bahl et al. (2016), and Fischer et al. (2017). The mechanism through which mindfulness could transform consumption behaviors begins with attention consumers learn to focus on their external stimuli and internal stimuli (body sensations, thoughts, feelings) and their effects on the consumption process and accept them with the attitude of nonjudgment, compassion, and an open mind, which help enhance awareness. Sustained awareness of the stimuli then facilitates insights into the true nature of any stimuli and self, which is always changing and impermanent. These insights can weaken consumers' attachments to habitual, unhealthy behaviors, detrimental to their well-being, in the long run, leading to transformative choices and experiences (Bahl et al. 2016). Bahl et al. (2016) reviewed empirical research demonstrating the effect of mindfulness on consumers' health, financial, and psychological well-being. Mindfulness enhances awareness of cognitive–behavioral processes underlying consumption that previously may have been relatively automatic, making people become more attentive to their own experiences and understand one's true needs and how one's actions affect well-being. Thus, mindful consumers will have more deliberate choices and are less susceptible to the persuasive influence of advertisers and others (Rosenberg 2004; Pollock et al. 1998). For example, Brewer et al. (2011) explained that mindfulness practice helps stop smoking through an attentional focus on immediate, embodied experience with an attitude of acceptance, which gives smokers an objective experience of what the smoking cravings feel like in their bodies and, if not acted upon, will diminish. Increased awareness also improves how one views oneself and sees how one is unfulfilled (Rosenberg 2004). Through some experiments, Brown and Ryan (2003) found mindfulness training can enhance one's self-esteem and satisfaction with one's own behavior, thus, reducing the likelihood of being motivated by social motives and increasing the capacity to find ways to fulfill psychological needs at a deeper and more meaningful level.

Despite these logical mechanisms to link mindfulness and transformative behaviors, the connection between mindfulness and consumer behaviors remains a largely

unresearched area (Fischer et al. 2017) and empirical research on the effects of mindful consumption is still missing, highlighting the research gap (Haider et al. 2022; Milne et al. 2020). While the extant empirical research found that mindfulness can reduce adverse effects of mindless consumption and enhance consumer well-being (Bahl et al. 2016), they employed experimental manipulations of mindfulness meditation and quantitative measures of mindfulness (dispositional and state measures) (Bahl et al. 2013; Brown and Ryan 2003; Brown et al. 2009; Hafenbrack et al. 2014; Hong et al. 2014; Van De Veer et al. 2016; see more in Bahl et al. 2016). Little has been known about how mindfulness-based intervention can transform consumption behaviors in practice and how the mindfulness mechanism can be implemented and sustained in daily lives amid many temptations, as being mindful takes time and effort to practice.

Feng et al. (2018) studied the difference between mindfulness in Western psychology using mindfulness scales and that in Buddhism. The former is conceptualized as non-judgmental, present-centered awareness, while the latter involves attention, flexibility, skillfulness, purposefulness, ethics, and wisdom. Buddhist mindfulness focuses not only on self but also on others and can be conceptualized as paying attention to what is occurring in one's immediate experience with care and discernment (Shapiro 2009). More importantly, Buddhist mindfulness is contextually independent and cannot be easily practiced in isolation from other interrelated concepts. That is, mindfulness is one of the parts of the Eightfold Path and all parts support and affect each other and have to be practiced altogether) (Rosch 2007; Lennerfors 2015). From the Buddhist perspective, being mindful enables one to realize that all physical and mental phenomena are suffering, impermanent, and no eternal self exists. Therefore, mindfulness in Buddhism aims at alleviating general suffering rather than serving as a therapeutic technique to remedy or prevent specific psychological disorders (Christopher et al. 2014; Rosch 2007).

To date, we lack understanding from active Buddhist practitioners who are regularly dedicated to Buddhist principles at the supramundane level, including but not limited to meditation practice. To our knowledge, none in the extant literature has delineated an in-depth account of the consumption experiences of active, dedicated Buddhist consumers and the process of transformation of their consumption behaviors over time. What obstacles they have encountered in the transformation process (Bahl et al. 2013), how they have gradually adjusted their consumption behavior, their dependence on material goods and the self (Bahl et al. 2016), and how they can appreciate consumption experiences without emotional attachment and ego defense (Mick 2017) are yet to be addressed. These led to our three main research questions about how active Buddhist customers practice and adapt consumption behaviors, the process and strategies of change, and the consequences of their consumption behaviors.

**RQ1.** *What are the consumption behavior patterns of active Buddhist consumers?*

**RQ2.** *How do they change their consumption behaviors?*

**RQ3.** *What is the mechanism underlying the behavioral change?*

Answers to these questions could provide more insights into how active Buddhist customers consume and manage their temptations and how we can nurture mindful consumers. This study would extend previous studies on Buddhist consumption, defined as the acquisition, usage, and disposition of goods and services to satisfy the desire for true well-being (Payutto 1994). Buddhist consumption centers on the middle pathway and avoids the extremes of self-denial and self-indulgence, which can harm physical and mental well-being (Puntasen 2004). The purpose of Buddhist consumption is, thus, not for pleasurable sensations or ego gratification but the physical maintenance required for intellectual and spiritual growth toward a noble life. Consumption is a means to an end, which is human potential development (Numkanisorn 2002; Payutto 1994; Zsolnai 2011).

Buddhist consumption shares a similarity with sustainable consumption, which ensures the satisfaction of basic needs and a better quality of life for the present and future

generations (Fischer et al. 2017; Haider et al. 2022). Both are concerned with adjusting the nature of consumption and minimizing disturbances in the natural world to deliver well-being and decrease suffering (Daniels 2011). Another consumer behavior related to Buddhist consumption is anti-consumption, a financially independent and intentional nonconsumption behavior by rejecting material goods or reusing already acquired goods (Cherrier 2009; Chatzidakis and Lee 2013). Unlike Buddhist consumption, anti-consumption is manifested through frugality and voluntary simplicity practices, creating a "resistance" identity (Cherrier 2009; Haider et al. 2022). From a Buddhist economics perspective, non-consumption may or may not lead to well-being. If it does not lead to well-being, it is mistreating the self. The question is not whether or not to consume but whether consumption leads to true well-being and human development (Zsolnai 2011). The challenge for consumers lies in how they can consume without being too attached to possessions to avoid suffering. This conundrum has yet to be investigated (Mick 2017).

Overall, in our review of Buddhism-rooted and mindfulness research works on consumer behaviors, there is a paucity of empirical research that links Buddhist principles and mindfulness with consumer behaviors (Armstrong 2021; Bahl et al. 2016; Mick 2017). Among a small but growing amount of empirical works, some (Kopalle et al. 2010; Pace 2013) focused on a specific Buddhism concept (e.g., Karma and Four Immeasurables), while others (e.g., Eckhardt 2011; Pongsakornrungsilp and Pusaksrikit 2011) examined Buddhist teachings and rituals (e.g., giving, praying, meditating). These works show the interconnection between materiality and spirituality, since lay Buddhists still display the self in their consumption and use materiality and consumption as a part of self-improvement. Further, the extant empirical research on mindfulness tends to employ laboratory studies, mindfulness intervention, and quantitative measures of mindfulness (see the summary in Bahl et al. 2016). While Milne et al. (2020) used a modern mixed method of text mining and clustering, they revealed three segments of mindful consumers who are mindful of price and quality tradeoff, mindful of firms' sustainability practice, and mindful of the entire production, consumption, and disposal process and consequences of the purchase. Their findings still offered a limited view of Buddhist mindful consumption. Given the difference between mindfulness and Buddhist mindfulness, the experience of active Buddhist consumers who regularly practice Buddhist principles has yet to be explored.

## 3. Methodology

The context chosen for this study is Thailand, a country with the World's second-largest number and percentage of Buddhists, especially for the Theravada branch, which is the oldest and most conservative tradition (Pew Research Center 2020). Thailand's sociocultural and economic system has been influenced by Buddhist values and Thailand's King Bhumibol Adulyadej's Sufficiency Economy Philosophy suggests a balanced and sustainable way of living, comprising moderation, reasonableness, and self-immunity principles, along with appropriate ethics and knowledge as supporting conditions (O'Sullivan and Pisalyaput 2015; Pusaksrikit et al. 2013). A sufficiency economy does not mean that one must constantly be frugal but suggests one should live a reasonably comfortable life, which can be extravagant occasionally if it is within one's capacity to establish economic stability. Sufficiency Economy emphasizes the Buddhist principle of the middle path, leading a person to be fully satisfied with what one has and is at peace with the self (Pusaksrikit et al. 2013). However, consumerism also challenges Buddhist values (Kraisornsuthasinee and Swierczek 2018). Thailand is one of the most popular tourist destinations in the world, where visitors and local people can enjoy good cuisine, sightseeing, and wellness indulgences (e.g., spa treatments). Thais also spend highly on food, traveling, health and beauty, and luxury items (Sachamuneewongse 2018), and, ironically, Buddhist-related goods and services (e.g., Buddha images, amulets, fortune telling) (Brox and Williams-Oerberg 2017). According to the Trade Policy and Strategy Office's survey in 2020, Thai people's annual spending on merit-making activities is estimated to be approximately THB 10 billion (Wornkorporn 2020).

This paper employed a qualitative approach to examine how engaging in Buddhist practice, which involves regular practices of concentration and mindfulness meditation, and meditation retreats change consumers' emotional responses and attitudes, affecting their behavioral responses. Mindfulness is a complicated, multifaceted concept and, more importantly, a highly personal and subjective experience (Grossman 2010). With such a complex and unobtrusive nature of these experiences, an in-depth interview is deemed appropriate to gain insights into and explanations of the experiences in a person's life in a subtle way (Belk et al. 2013). To understand the practice and consumption experience and behavior change in active Buddhist consumers, we recruited active Buddhist practitioners through both purposive and snowball sampling. Snowball sampling is a method that considers a participant's social networks, helping to reach a specific group of people who are hard to get (Browne 2005). Purposeful sampling was also adopted in identifying and selecting the information-rich cases (Patton 2002) from individuals who are knowledgeable about this Buddhist practice experience, willing to share, and able to articulate their personal and subtle experiences (Bernard 2002). Since researchers have been interested in this topic before the paper was initiated, we have acquaintances who have been dedicated to Buddhist practice for years. Some of them also recommended their friends who meet the criteria: regularly practice (by reading books, listening to Dhamma talks, mediating formally and informally in their daily activities, attending retreats, volunteering at temples, and consulting their Dhamma teacher regarding their spiritual progress) and observe significant changes in their consumption behaviors and lifestyle. In addition to the interview, a few interviewees documented their experiences in a reflective diary, which served as a complimentary data source.

This study reports in-depth interviews with 15 active Buddhist practitioners who lived in the city with an urban lifestyle. The sample size is guided by prior qualitative interview studies on mindfulness-related topics that interviewed 10–14 people (Böhme et al. 2018; Christopher et al. 2014; Eberth et al. 2019; Frank et al. 2019; Langdon et al. 2011; Stanszus et al. 2019; and Von Essen and Mårtensson 2014). A profile of interviewees is summarized in Table 2. The interviewees' names are fictitious. Our interviewees represent the age group between 24 to 55 years who have practiced from 1 to 17 years. A diverse range of experiences can provide both the recency and breadth of the experience. We believe that recent practitioners can still remember their experiences before practicing better than long-time practitioners who can provide more details about the change processes. Their occupations include student, government officer, employee, self-employed, and housewife. Most interviewees are Thai except two from Europe who have been interested in Buddhism, attending meditation courses in India and Thailand, and one graduating in Buddhist studies in Thailand. All interviews were conducted face to face in Thailand by either one of the authors and lasted between one and three hours. Some interviewees were interviewed more than once. The initial rounds of interviews provided the background of interviewees, their practice motivation and goal, and their past and current consumption behaviors. Subsequent interviews seek to explore more how their consumption patterns are transformed, what is required of them, and what challenges they have faced.

**Table 2.** Interviewees' Profile.

| Name | Gender | Age | Ethnics | Occupation | Years of Practice |
|------|--------|-----|---------|------------|-------------------|
| Ball | Male | 35 | Asian | Employee | 7 |
| Kanya | Female | 33 | Asian | Employee | 2 |
| Mook | Female | 37 | Asian | Government Officer | 2 |
| Jae | Female | 48 | Asian | Self employed | 6 |
| Aim | Male | 36 | Asian | Government Officer | 12 |
| Anna | Female | 24 | Western | PhD Student | 5 |
| Yui | Female | 37 | Asian | Self employed | 4 |
| Gate | Male | 50 | Western | Employee | 7 |
| Koi | Female | 38 | Asian | Housewife | 1 |
| Tara | Female | 40 | Asian | Housewife | 3 |
| Opas | Male | 45 | Asian | Employee | 10 |
| Kloy | Female | 36 | Asian | Self employed | 17 |
| Nitcha | Famale | 55 | Asian | Self employed | 2 |
| Nan | Female | 34 | Asian | Government Officer | 7 |
| Primprao | Famale | 33 | Asian | Employee | 4 |

Data analysis occurred during and after data collection to capitalize on opportunities to follow up on insights before the completion of interviews. Data gathered from the preliminary interviews guide the direction of subsequent interviews and the selection of interviewees; thus, the collection and analysis of the data are interrelated (Leavey et al. 2007). Interview data were recorded, transcribed, read through, and analyzed using two approaches. As Milne et al. (2020) emphasized, the consumer perspective of the role of mindfulness is understudied. An inductive approach can provide rich information that enhances our understanding of mindful consumption at this early theoretical development (Langdon et al. 2011). Thus, we began with an inductive approach by assigning interpretation to the transcript in an emergent fashion, identifying recurring words and concepts, rearranging them into categories, and discussing among authors key patterns across the interviews (Strauss and Corbin 1998).

However, to explore and understand the stage of behavior change for active Buddhist practitioners, we also looked at existing models describing stages of behavioral change as a guideline. The most famous one is the Transtheoretical Model (TTM), developed to help people change their undesirable behaviors (Prochaska and DiClemente 1984; Prochaska et al. 1992). The TTM posits that people do not change behaviors quickly and decisively. However, change in habitual behavior occurs continuously through a series of five stages, whereby people can recycle through the stages or regress to earlier stages from later ones. These include (1) pre-contemplation (not intending to change behavior), (2) contemplation (aware of benefits of change and costs of change), (3) preparation (make plans and ready to take action), (4) action (have made overt changes), and (5) maintenance (sustain their behavior change and work to prevent relapse to earlier stages). The other model is the Stepwise Model of Behavioral Change (SMBC), which aims to break the unwholesome habits and cultivate desired habits for a successful behavioral change (Dahlstrand and Biel 1997). This model comprises seven stages: (1) activation of the value of the new behavior, (2) attending to the present behavior so as not to automatically behave in a habitual manner, (3) considering alternative solutions in achieving the new behavior, (4) planning new behavior, (5) testing new behavior, (6) evaluating new behavior in terms of monetary, physical, and psychological costs, and (7) establishing a new habit through repeated reinforcement or old behavior remains intact.

In applying the behavior change model, the Buddhist practice experience differs from other problems to which the TTM has been applied as there is no single desirable target behavior for change when studying the experience of active Buddhists. The SMBC model developed to cultivate desirable habits is an alternative. We developed a set of codes guided by and applied from the TTM and SMBC model as well as a journey of mindfulness from Langdon et al. (2011). The approach is consistent with qualitative abduction, which

facilitates the discovery of new concepts through a combination of empirical facts with previous theoretical knowledge (Baron et al. 2006).

## 4. Findings

The data are structured into three sections according to the research questions: current consumption behavior patterns, stages of behavior change, and the process of change. The first section describes why, what, and how our interviewees consume in contrast to the past. The second section describes how our interviewees adapt to deal with temptation through a series of stages of change. The last section summarizes the process of change and strategies that facilitated the movement through the stages of change to achieve mindful consumption.

### 4.1. Consumption Behavior Patterns

Interviewees report that regular and consistent Buddhist practice has transformed their lifestyle and lifetime goal. For those who have recently practiced, their goals in life change from being wealthy and successful to merely having enough to eat, sleep well, and have good health, as they learned that even successful, rich, and famous people could not avoid suffering. Those who have practiced for several years expect not only to live simply and peacefully in a chaotic world, but also to be liberated from suffering and enlightened in their future lives. Since their attitudes, lifestyles, and lifetime goals change, their consumption motives change, affecting why, what, and how they consume.

***Why we consume.*** People consume when they recognize they have certain needs. Maslow's hierarchy of needs suggested various levels, from physiological and safety needs to social, esteem, and self-actualization needs. Our interviewees focus on the first two levels that satisfy basic needs and downplay higher levels of needs. In the past, they consumed goods and services to enjoy sensory experiences (hedonic consumption), to display and enhance their status (conspicuous consumption), and to offset their unfortunate events in life (compensatory consumption). Today, their consumption is motivated by basic needs, beyond which consumption is less likely to occur, as seen in the following comments.

> Opas: "After being ordained, I realize that we consume four necessities beyond what is needed. In the past, I bought clothes that have never been used. I spent millions decorating a house while I spent more time at the office. I enjoyed eating to satisfy my desires rather than hunger, resulting in being overweight."

> Ball: "In the past, when I felt stressed, I drank to escape from problems. I sought a delicious dish to enjoy and compensate for my daily tiredness. I wore a gold necklace to show my success and a batch ring to show my relationship with my classmate. I bought an iPhone and Starbucks as symbols of the modern lifestyle. Now I realize that these do not help improve my life or achieve my long-term goal, so I eliminate them. I want to have enough food to survive and live in a small house but safe enough to protect me. I no longer traded stock daily to become rich. I change to a value investor and trade for financial security."

Interviewees realize they have consumed more than their life needs and problems exist when they consume more than they need. Through some practice or self-experiment, they learn that they can live their life comfortably enough, even though they sacrifice some conveniences and pleasure from extravagant consumption. Further, they do not seek products to feel pleased, show their status, and escape from reality, as they realize products cannot solve their problems and only temporarily make them feel better.

***What we consume.*** People consume goods, services, experiences, symbols, and stories. In consumerism, people consume these beyond their functions (utilitarian value). They consume to feel good, to satisfy the five senses (hedonic value), and to express and enhance the self (symbolic value) (Holbrook and Hirschman 1982). Active Buddhist practitioners we interview live a simple life with fewer non-essential goods and emphasize the functional values rather than hedonic or symbolic values of products. When they practice seriously, their craving for consuming products and experiences declines and, thus, they no longer

engage in activities that cannot help them liberate themselves from suffering. They found those activities a futile waste of time.

> *Ball: "When my iPhone was out of order, I went to the store asking for any brand I could call and surf the Internet, and I chose the reasonably priced one. I ignore extra features and brands. I realize that brand-name is not real and meaningless. I visited the massage house less than before and resorted to Qigong which gives me good health, not temporary comfort. Before, I traveled with my friends every month to have fun. After practice for years, the desire for such activities disappears. Today, I don't feel like traveling, going to karaoke/parties, or participating in other activities that cannot help me liberate myself. I donated comic books I collected. I found playing Facebook and online communities is a futile waste of my time, better spend on meditating."*

While Buddhists are often portrayed as avoiding consumption and rejecting the material world, Buddhist mindfulness is ethically wholesome and involves compassion for all living beings (Feng et al. 2018) and the consumption and non-consumption of active Buddhist practitioners also depend on impacts on oneself and others. Some interviewees said that sometimes they bought things not necessary to them if the purchase can help others.

> *Yui: "I have stopped eating meats after practice. I felt bad that we let tastiness take away animals' lives. My lifestyle changed; I frequently visited temples instead of nightclubs. I observe monks' simple life, so my fashion style changes to be minimal; no make-up and hair dye. I became less attached to the brand name. I can use any brand that is reasonably priced and of moderate quality. Further, I'd like to support products sold for charity purposes, although I may not need such products."*

***How we consume.*** Practicing also changes the way they shop and consume. A consumer decision-making journey comprising five steps is adopted. In the first step, they only consider a purchase when it is a need/problem (e.g., products are used up or out of order), not a want/an opportunity (e.g., a new model is launched). They are less susceptible to marketing and social influence. Second, they would search for information to aid an acquisition decision and keep it to a minimum, rather than enjoy regularly searching to stay up to date on new products. Third, evaluation among alternatives is based primarily on the functional and long-term benefits of a product, price and quality, and durability. They rely on cognitive deliberation rather than emotion. Fourth, making a purchase becomes a simple task. They create a plan before shopping and, thus, spend less time, are less sensitive to in-store promotion, and are less impulsive. The amount of purchase is also changed. Some interviewees mentioned they no longer buy the same product in multiple units for stocking up or buy bags/shoes in various colors to match the colors of clothes. They realize that they can only use one at a time. Many items they have bought before are unused and become spoiled or damaged. For frequently used goods, they use the same brand or any brand available in the store and have similar functions and prices. In the consumption stage, they are less likely to cling to the experience of consuming a product. They can enjoy a product but to a lesser degree and only at the point of consumption, not keep thinking about it afterward (i.e., become neutral towards the product and less attached to a brand). If they are unsatisfied with the purchase, they just accept it and do not seek complaint. For a satisfying purchase, they would use a product until they cannot use it. If the product no longer performed well, they would buy a new one without whining or clinging. Clinging habits during consumption and post-consumption can be seen in the following excerpt.

> *Ball: "When I eat or use products, I no longer cling to them. Before, if I enjoyed a dish/movie, I kept thinking of it, revisited the restaurant/theater, and wrote an online review to share with others. Now, I no longer enjoy searching for restaurants. For food I like, I still feel that it tastes good, but I enjoy it to a lesser degree and only at the time I am eating. After that, I don't think about them. Only focus on the current moment."*

*4.2. Stages of Behavioral Change*

Based on the interview data, the interviewees underwent several stages before their behaviors changed. Their stages of change extracted from the interview do not strictly follow the current behavior change models and, thus, the alternative model was proposed and adapted from the existing models. Now that we understand their behavior changes, the next question is: how do these changes take place?

***Stage 1: Problem Recognition*** is the first stage, in which interviewees recognize that their problems cannot be solved by consumption and, thus, resort to Buddhist practice as an alternative solution. When experiencing suffering, interviewees often go shopping or consume some products to feel better. Still, such feelings are short lived and soon they suffered again, went shopping, and consumed again, leading to further suffering again. Hence, they realized they needed an alternative to consumption. While some resort to Dhamma practice to help them go through suffering in their lives, such as illness and failure in their career, others have a great life but still feel empty inside. They questioned why they are still not that happy when they have everything (money, success, and love). They tried to search for what could bring them happiness and fulfill their purpose in life and they were recommended by their friends by a Dhamma book to learn and practice Dhamma. What is different from other self-control strategies is that consumers are interested in the Buddhist practice for a high-level goal in life rather than solving specific consumption problems, such as obesity, compulsive buying disorder, and debt, which are often consequences of their life problems.

***Stage 2: Trial and Error*** is the stage in which interviewees are aware of the potential path to a better life through Buddhist practice and decide to try it. Buddhist beginners can practice at the temples or religious schools by participating in short or long meditation programs or live-in meditation retreats. There are three primary practices: following five precepts or moral conduct (*śīla*), practicing concentration meditation (*śamatha*), and insight/mindfulness mediation (*vipaśyanā*). These involve physical, mental, and wisdom training.

***Five precept practice:*** without a moral life, it is impossible to succeed in concentration and insight meditation. Some interviewees described how they trained in morality and experienced positive changes, typically from abstaining from gross defilement, such as alcohol consumption or buying animal skin clothes.

> Ball: *"I started practicing after reading the book "It is a pity the dead didn't read this book", which made me question the purpose of life. The book challenged me to follow at least one of the 5 precepts for 3 months and see the result myself. After I tried to stop drinking, I found myself smarter and sleep well than before".*

***Concentration meditation*** involves training one's mind to be calm, to focus on one point to bring about a state of consciousness. Concentration techniques popular in Thailand include concentrating on either in-and-out breath, the rise and fall of the abdomen, or the word 'Bud-Dho.' Some experience a state of calmness sometimes, other times, mind wandering, boredom, or drowsiness.

> Tara: *"When I practiced meditation, I learned to focus my attention on my breath. It was not easy and not fun to pay attention to only the breath. My mind often wandered, and it had me think about many things (good, bad, neutral) from the past or future, even something I never thought I would think about, like nuclear. It was scary that my mind never stopped thinking; many are evil thoughts. I had to tell it to stop and to focus on the breath again".*

***Insight meditation*** involves interviewees learning to understand the characteristics of physical and mental phenomena (impermanent, suffering, not self). When we concentrate on bodily and mental states, we can see that any phenomenon, such as feeling numb, hot, calm, or mind-wandering, arises and fades away.

> Kanya: *"When I practice Vipassana, I observe motions on different body parts like a heartbeat and sweat. It was fascinating to see that different body parts elicit different*

*feelings (e.g., painful, itchy, burning, electric feel), and even some areas like the elbow can feel some motions though very light. The teacher told us that for each thought, feeling, or body sensation that arises, just observe, not meddle with it. We could see it fades away, arise again, and fades again. I wondered how my numb feeling would disappear if I didn't change my posture. To my surprise, when I tried to bear the pain patiently, the feeling's actually gone."*

Some people practice morality and meditation at retreats, where they are away from home for several days and must follow the rules, such as practicing five precepts, eating whatever is provided (usually vegetarian) only twice a day, not speaking with anyone, not using any electronic devices, cleaning the monastery/toilet, waking up very early, and meditating and praying for 12–14 h a day. This could be considered physical exercise. Anna described her experience as follows.

*Anna: "When I went on a retreat in India for two weeks, on the first day, I felt like I was in prison. I felt regretful and scared. I had to sleep with others and take a bath in cool water, an experience that was not comfortable. Meditating the whole day was exhausted and painful. Sometimes, I was not allowed to change my posture. I felt itchy, painful, numb, and upset. I was confused about what I could learn, but I could not go back. I wanted to know what happened at the end. After two days, I got used to the new environment. I learned I'm fine even though I eat two meals, sleep on a hard mattress, and cannot use the Internet. It was a great discovery that my life does not need many things, but I can live happily."*

Overall, at this stage, interviewees reported some positive changes after practice, including feeling calmer, able to control their emotions, and increased awareness helps them better solve their problems in life. However, they experienced physical and mental difficulties before they got used to the practice and experience peace of mind. For effects on consumption after practice, interviewees began to recognize that their consumption level is sometimes higher than necessary but that they must control and restrain themselves from consuming in ways that do not contradict Buddhist values (e.g., five precepts), which is quite difficult to do at this early stage.

***Stage 3: Determination and Obstacles***. At this stage, interviewees intend to pursue their practice more seriously. They seek to listen to Dhamma teachings and participate in Dhamma seminars or gatherings during their free time. Some plan to follow the precepts strictly to reduce further gross defilements and they start to see themselves as a morally good person. The consumption that violates the five precepts is decreased further.

*Ball: "I decided to really follow the precepts when I found out that giving up drinking is good for my health. In the first year, I did fewer mean things, like kill fewer mosquitoes and eat less meat. I stop doing all of these things in the second year. I feel I am kinder and happier. Practicing changed the way my mind worked, making it more moral and clean."*

As they spend more time on Dhamma-related activities, this reduces their time for entertainment activities and demand for products, such as fashion, which is not needed when they participate in Buddhist activities. They better control their temptations by spending more time deliberating on consumption choice, comparing the price/quality of alternatives, and evaluating the purchase necessity, urgency, and impact on their well-being. At this stage, they still frequently give in to temptations, even if they know that the purchase is unnecessary and not consistent with Buddhist principles.

During this stage, the interviewees also reported problems they faced when practicing and foregoing consumption in their daily lives. These include the unpleasant feeling that the desire is not fulfilled and the social pressure from others. For example, *Yui* experienced conflict since she refused to attend the social gathering and her friends referred to her as a nun.

*Ball: "It is not easy to practice; I have to control my mind and action when I see my favorite food. It was awkward telling my friends that I would no longer be drinking with them. At first, they tried to get me to drink, but I said no. Then, sometimes, they put*

*social pressure on me, which hurts. Finally, I stopped hanging out with those friends and joined a group of friends who enjoyed being outside in nature."*

Another obstacle to practice includes busyness and laziness. *Kanya* joined a 6-month meditation program where she learned Dhamma teaching, meditated, and helped out with volunteer work every day for six months. Still, after the program finished, she was distracted by her work and laziness and resumed her normal life.

*Stage 4: Progress and Mistakes.* At this stage, interviewees report more progress in their practice and behavior changes than in stage 3. They are committed to daily practice and schedule it as a higher priority than working out or socializing with friends/family. They take more advanced Dhamma courses and experiment with various practice techniques. Some suggest that they progress significantly after finding the practice techniques that suit them the most. For example, *Primprao* mentioned that she could practice daily after taking the 10-day course from one school that emphasizes Vipassana practice and limits Buddhist rituals common in other schools. Similarly, *Kanya* practiced more effectively when she learned from a training school to concentrate on her body sensations, as she is quite sensitive to her body conditions. Unlike her husband, who is a thinker, he would benefit more from mindfulness in his state of mind. *Koi* mentioned that she tends to fall asleep with sitting meditation and cannot concentrate with walking meditation, but she gains concentration when running on a treadmill.

At this stage, interviewees witness gradual progress in managing gross and subtle defilements. They seek to refrain from such actions as using a company's resources for personal use and listening to music that may contradict the second and seventh precepts. As a result of their improved ability to adhere to the precepts in a more refined manner, their consumption of items in conflict with Buddhist values is significantly reduced without exerting as much self-control. Nonetheless, they sometimes yield to their temptations or social influence, resulting in feelings of guilt mixed with anger for making mistakes before they tried to deal with guilt through self-compassion and acceptance. For example:

*Primproa: "I was recently shopping at a mall and walked past a beautiful dress that matched my taste. Even though something in my head said it was unnecessary, I tried to come up with a counterargument that it could be used in the future, and I finally bought the dress with an initial sense of satisfaction. Later on, a sense of guilt emerged. 2000 THB I spent on the dress would be enough to buy a necessity like food for several days. I became angry at myself. Why did I buy it? Where has my minimalist style gone? This dress is unnecessary, and I can easily rent designer clothes these days. In the end, since I can't return the dress I purchased, I must forgive myself and move on, promising not to succumb to the temptation next time."*

Further, at this stage, interviewees encounter some obstacles due to the complication of more advanced practice in their life. For example, *Ball* mentioned how difficult it was when he tried to observe the eight precepts in his daily life requiring him not to eat after noon and enjoy the entertainment. Yet, his company often held dinner meetings with colleagues and customers at nightclubs, which played loud music, disturbing the peace he valued. Like stage 3, some experience life stage changes, such as getting pregnant and entering college, affecting their ability to practice and consequent consumption behaviors. *Nan* had meditated since she was 10 years old. Until 18 years, when she got into university, she had to adapt to the new environment and could not meditate as much as before. She also had to follow the trending lifestyle of university students.

*Stage 5: Insight and Transformation* is the stage in which interviewees are more advanced in their mindfulness practice. They are mindful of what they think, how they feel, and what they do in their daily life. This increased awareness enables them to observe the true nature of things and gain insight into their true needs that facilitate optimal consumption levels. Some interviewees shared the experience of their mindful eating, which helps them locate the point of satiety, realize their true sense of taste, and remind them of the purpose of eating and eating with a purpose (e.g., for energizing, for good health).

> *Opas: "I love eating and always eat two dishes a meal before. As I have practiced being mindful of physical activity and mental formations, I was aware of the taste of the food for the first bite, the second bite, and so on of which the flavor goes dull, aware of where I sense the taste, aware that I was swallowing rice, and aware of changes in my body such as stomach expanded and the feeling of fullness, and thus no need for the second dish."*

While in stage 4, they may be required to exercise some control over purchases of products that are not inconsistent with Buddhist principles but are unnecessary and induce pleasure. At this stage, their consumption patterns become increasingly refined. Being mindful of their needs, well-being, and Buddhist principles becomes their natural habit.

To sustain and continuously progress the practice, interviewees contemplated and decided to change their lifestyle, hobbies, and even occupations to be conducive to Buddhist practice. For example, *Ball* left his demanding full-time job to work as a freelance Math tutor for students. *Kloy* became an online seller so she could have time to volunteer at a temple. *Yui* changed her job to be a small entrepreneur selling food and drink and she frequently provided free food for the poor or local heroes. Her lifestyle changed significantly from a frequent nightclubber to a part-time rescue volunteer and meditation training assistant. The new occupation and lifestyle may not provide high earnings and professional success. Still, they fit well with the interviewee's priority to practice, help them understand the value of human life and support others, and provide them with emotional comfort, so they do not require consumption therapy. Along with the lifestyle change, the most noticeable difference in some interviewees, such as *Ball, Aim, and Yui*, as observed by their acquaintances, is their appearance and personality. For example, *Yui* transformed from a woman with a cheerful, funny, outspoken personality who enjoys dressing up in colorful fashion and hairstyles to a woman with a calm and kind nature and a plain and minimal dressing style. She speaks softly and moves slowly and with discretion. The change in their personality and lifestyle further influences their consumption and shopping behaviors.

*Stage 6: Tight and Tear.* Following their mindfulness practice and skill and transformation of their way of life and consumption habits in stage 5, interviewees made significant progress in their Buddhist practice and subsequent consumption behaviors. However, some became so attached to their mindfulness and spiritual advancement that they engaged in an even stricter Buddhist practice to advance faster on their spiritual path (i.e., being a *Sotapanna* who entered the stream leading to Nibbana, the state of total liberation from the cycle of rebirth and death, the ultimate goal of active Buddhists). However, a more stringent practice that determines how they live their lives and consume goods and services leads to frustration, emotional discomfort, and difficulties in living their life and living with others in the world. For example, Kloy recalled a frustrating time with her friends when they were still interested in nonsense or entertainment topics.

> *Ball: "While trying to follow the 8 precepts every day, I was overly strict with myself, particularly regarding what I eat. I won't eat Chinese food that uses XO (brandy) sauce. I became a vegetarian and thought that meat eaters were bad people. I even told a meat-eating friend that they had body odor. I was not at ease with those around me who did not adhere to Buddhist precepts. I also force myself to move slowly, eat and chew slowly, and refrain from listening to music that does not help my practice progress. I felt terrible about myself when I couldn't control myself and gave in to a food craving."*

In terms of consumer behavior, interviewees report minimal consumption, as in the previous stage, but with the addition of restraint behaviors to remain mindful at all times, as well as the complexity of the consumption process, which must strictly adhere to the more advanced Buddhist principles. Such behavior causes a restless mind during consumption and feelings of guilt and shame if they succumb to temptation. Interviewees recalled how difficult their lives were back then. Finally, they realize this is not the proper practice and try to strike a balance so they can live in peace in both the material and Dhamma worlds.

*Stage 7: Balance and Freedom.* At this stage, interviewees have found the right balance between Buddhist practice and consumption and they no longer feel constrained by temptations in consumption or religious principles. Those who reach this stage understand

that they cannot simply avoid consumption. They still consume products and services to satisfy their needs (not wants) and improve their well-being. Their consumption behaviors are simple, functional, and purposeful and they can still live in peace with others in the consumer society.

> *Kloy: "Buddha did not teach us not to consume but rather to consume with valid reasons and in suitable circumstances. As a woman, it is difficult to be ordained, I still have to live in this world, and consumption is still necessary. I still go shopping when I need certain products to live my life or work. I can go into the store if there is a sale, but whether I buy or not depends on my needs. If there is a chance, I can still eat well, but I no longer yearn for it or feel guilty about taking or not taking it."*

At this stage, their desires for unnecessary goods or services are reduced significantly (some desires are even extinguished), even though they do not exert some control strategies. Through consistent Buddhist mindfulness practices, interviewees understand the impermanent nature of desire and suffering. When they are aware of their desire, they see that the desire fades away by itself as other thoughts or feelings emerge and replace them. In previous stages, they still have to exercise some control to resist temptation, but at this stage, they do not need to resist temptation. If the temptation arises, they are aware and observe the temptation until it disappears. Sometimes they may give in to trivial temptations but no longer feel guilty. They are also less attached to the pleasant or unpleasant feelings involved before, during, or after consumption or non-consumption. As they focus on the present moment while consuming goods or services, they can still have a brief moment of pleasure, but this pleasure is fleeting.

> *Ball: "I now understand the middle path. When I am hungry, I eat but do not eat for meaning. I can eat without counting pieces and calories and stop when full. If friends invite me to join a trip, I feel neutral between going and not going. I may join the trip if I think I can learn new good experiences and the budget permits. I did not cause trouble for anyone. I still like some food, but I no longer feel aroused by it. Earlier, if I had a desire for a certain food, I had to control such desire. Now I feel free. I do not need to control it. I can see the desire comes and goes. I do not feel positive when fulfilling it or negative when I do not."*

In the last stage, interviewees realized the true nature of Buddhist practice. They reached the stage when they did not have to control their desires because it is not for control but understanding, accepting, and letting go.

### 4.3. Process of Change

After we understand the stages of behavior change in active Buddhists, the question remains the process that explains the activities that active Buddhists use to progress through the stages and help them make and maintain changes toward being less dependent on desires and free from the consumerist chain. To progress through the stages of change, active Buddhists can apply cognitive, affective, and conative processes. The following processes of change and tactics have been identified, with some being more relevant to a specific stage of change than other processes (see Table 3). These involve three modes of existence (having, doing, and being), the manner in which the individual reacts to stimuli's outer or inner world (Fromm 1976; Rand 1993).

**Table 3.** The Process of Behavior Changes.

| Mode of Existence | Mechanism | Strategy | Behavioral Outcome |
|---|---|---|---|
| • Doing<br>• Having<br>• Being | • Stay calm<br>• Aware of cost-benefit<br>• Create inner happiness<br>• Understand the reality of self and suffering | • Detachment<br>• Reminiscence and Projection<br>• Conducive Environment<br>• Merit making<br>• Decomposition | • Less sensitive<br>• Independent<br>• Sufficient<br>• Self-fulfilled<br>• Less attached |

***Stay calm:*** Numerous studies (e.g., [Baumeister 1997](); [Tice and Bratslavsky 2000]()) suggested that strong emotions, especially negative ones, can contribute to self-control failures because, to feel better from a bad mood, people might seek to indulge themselves by buying or consuming goods/services. Consistent with a growing body of scientific research demonstrating the beneficial effects of mindfulness on emotional regulation ([Lutz et al. 2008](); [Chambers et al. 2009]()), interviewees report that their emotional intelligence is enhanced after practice. Without overt control, regular concentration meditation has transformed interviewees to be calm, even tempered, and emotionally stable, from both their own and others' perspectives.

> *Jae: "In the past, when I argued with my boss, I went shopping and bought expensive things like an expensive diamond, Bose, Karen Millen, I regret later. After regular concentration meditation, I controlled anger better before. Since my mood is stable, I no longer shopped as retail therapy."*

One strategy interviewees use to maintain their emotions is ***Detachment from emotion-induced activity***. Mook and Ball said they avoid activities (e.g., listening to music, watching TV/movies, and monitoring up-to-the-minute stock index) that can lead to mood swings and affect their sense of judgment.

> *Mook: "I stop listening to music as it can arouse my emotions. Listening to songs makes me tired, hindering me from achieving my meditation goal. Often, when we listen to happy or sad songs, we become daydreaming or depressed even when we have no experiences as in the song. Today, I listen to Dhamma talk instead."*

After regular Dhamma practice, interviewees reported that their feelings for some products faded away. In the initial period, interviewees are mindful, deliberate a while before purchase, and curb unnecessary purchases. Over time, these significantly reduced desires and cravings for products without deliberate control. In addition, interviewees reported feeling less sensitive to marketing stimuli, such as new products and sales signs, as meditation makes them calmer and less attached.

*Aware of two sides of the coin:* When we stay calm, our mind is stable and we can see our thoughts clearly, like sediments in still water. As a result, active Buddhists can be aware of both benefits and risks or negative aspects associated with consumption and become less likely to fall prey to the illusion that consumerism can offer consumers anything they want ([Miles 1998]()).

> *Gate: "It is like when we are young, we play with sands and rocks, when we grow up, we learn that these are bad and we stop playing with them. If we are not mindful, we are blind to the risks or harms consumerism can cause. For example, behind good taste, if we over-consume, we will have health problems. After regular practice, I have learned to leverage reasoning to overcome desires."*

Consumers usually think that they will be happy when they have products. Nonetheless, attaching to happiness from goods affects peace of mind and creates mental unrest and, thus, it becomes unhappiness. Interviewees report, ***recalling from past experiences,*** a feeling of anxiety over how to acquire, consume, maintain, and dispose of products.

> *Nitcha: "If you are mindful, you will realize that buying things does not make you happy. When I was younger, I wanted a brand-name bag like my friends'. I was anxious as I looked for the right one and kept up with new collections. When I used my first LV bag, I worried it would get dirty, damaged, or lost. I always put the pad there to keep my bag from getting dirty. I had to put it in a plastic bag when it rained to avoid stains. I have hurt again when the bag I hardly ever used went out of style."*

> *Kloy: "I remember when I was studying in the United States and had to bring a lot of stuff back home." I felt burdened and exhausted. Many items still carry a price tag. Some of them were worn but never used. Products, like humans, can grow old and die. Today, when I buy new items, I use them rather than keep them, and I only buy when necessary."*

Another strategy is projecting the potential impacts of consumption in the future, as one interviewee explained.

*Koi: "Recently, my mom told me someone wants to sell a husky puppy. I imagine how cute it is for seconds, then I think about how I have to spend time playing with it, taking care of its food and poo, and feel sad when it dies. I decide not to adopt it."*

Being constantly aware of potential risks could help consumers to become less dependent on consumption. They may also gradually change their attitude in their memory (*sañña*) regarding the attractiveness of products and, therefore, craving for products may be reduced little by little.

***Outer happiness did not last long:*** Consumption may increase happiness as it reduces hardship, provides comfort and pleasure, increases status, and increases social connectedness ([DeLeire and Kalil 2010](#)). However, interviewees realize that happiness from consumption is impermanent, insatiable, and unsaturated.

*Opas: "Once we own a product, later we get bored, we want a new one, we get bored again, and want a new one again and again. So why do we struggle to get a product in the first place?"*

*Gate: "I consume fewer goods and services because happiness from consumption fades away quickly, and I need to consume again to be happy. Now I found authentic happiness. Dhamma practice brings real, not shallow happiness obtained from consumption. It is light, not extreme, not exciting but could fulfill my mind and lasts longer."*

***Inner happiness is light but lasting***: For active Buddhists, their happiness depends less on consumption. Happiness lies not in the possession of products, but happiness is created from nature and within. Interviewees mentioned that when they connect to nature and meditate, they feel peaceful and self-sufficient. There is no need to pursue happiness from the outside.

*Anna: "I'm now easily satisfied. I can feel happy when meditating, viewing the beauty of the sky, enjoying the breeze, and seeing the leaves dance."*

*Ball: "When I meditate, I feel deeply peaceful and relaxed, so now I don't have to go outside shopping or hang out with friends to be happy. I just stay at home, sit back and relax with meditation, and I'm happy. Today, I'm happier when meditating than traveling and watching movies, so I have stopped thinking of those activities. Meditation fulfills my mind. It's like when we're full, we can't eat anymore. I can't think of other things when my mind is full."*

An effective strategy that can aid in creating and maintaining happiness is creating an environment ***conducive to inner-peace development***. This strategy includes altering the home environment, way of life, occupation, and social network.

***Understand the reality of self***: Dhamma practice reduces 'self' or 'ego', as opposed to consumerism, which emphasizes it ([Mick 2017](#)). Through consistent Buddhist practice, they learn to think, speak, and act positively and feel that they have become better people as good Buddhists. They understand their own value and self-worth; thus, they no longer require products to increase their intrinsic value. Consequently, the symbolic value of products is diminished.

*Mook: "Practicing Dhamma helped reduce my ego. I care less about what others people think of me. Just doing good things is enough. My mind has been uplifted. I become kind-hearted and ashamed of sin. My internal value is enhanced when I feel I'm a good person. I do not need to use a high-end brand to show my value. Those products are meaningless; they cannot enhance my value."*

Interviewees reported that engaging in Buddhist ***merit-making activities,*** e.g., donation and retreat, helps them feel that they are no different than others. People are born, suffer, and die, regardless of who they are. They become kind and show compassion towards others more. As they give more to others, their attachment to self is reduced. They learn that humans suffer because of who we are. Attaching to self is the origin of suffering

> *Koi: "We learned in a seminar, and I tried to remind myself about this: when we don't mark a product as 'My belongings,' we don't suffer when a product changes its condition (old, worn, stop functioning). We become neutral when a product rather than our product gets lost."*

When they realize that 'self,' 'I,' 'my' is trivial, interviewees cannot find valid reasons to consume goods and services to be unique or different from others.

> *Kloy: "When we use luxury goods, we can see that we feel proud, we become cheerful, but what's the point of showing off when others would not feel like you? Showing yourself is meaningless."*

> *Aim: "In the past, I was complimented when using perfume. I felt more confident and loved, so I became addicted to perfume. I want others to think of good smell when they think of me. Now self-identity is not important. I realize that confidence is created from within, not dependent on materials. Therefore, the role of perfume disappeared."*

Furthermore, some interviewees describe the process by which they learn to understand the truth of phenomena and self, which is composed of Five Aggregates (*skandhas*): Consciousness (*viññāṇa*), Physical form (*rūpa*), Mental formation (*saṅkhāra*), Perception/ Memory (*saññā*), and Sensory experience (*vedanā*) (Nyantiloka 1946). A combination of these aggregates (five *skandhas*) form who we are (i.e., 'I,' 'self,' or 'identity'). In Buddhism, the self is a delusional formation that represents the outcome of our never-ending chain of cravings (Brazier 2003). These five aggregates, when considered together as a collective whole, can cause suffering. However, when viewed separately, their power wanes. Often, we react habitually because of our past experiences and, thus, to eliminate bad habits, we need to understand and decompose elements of body and mind.

> *Koi: "I attended a one-month 'how to be a human' course. We learned the process of how suffering begins and ends and practiced deconstructing five aggregates in our daily lives. In a class, I saw (conscious) someone's polka dot 'Kate Spade' bag (physical form). I recalled that it is a polka dot, and I always like anything polka dot (memory) because it is cute (perception). Seeing it made me happy (feeling), and I thought it would be nice to own it (mental formation). Without the memory "I always like polka dot," I don't think I like that polka dot bag. That is Aha! Moment, I felt enlightened. When I saw a polka dot on any object later, I felt I did not have to like every polka dot, and I didn't like them as much as before."*

Through these processes and strategies, interviewees have experienced a simpler, more peaceful life, with less anxiety. For example, by buying fewer clothes and donating unused ones, *Yui* found it easy to choose how to dress in the morning. *Nitcha* can attend a wedding ceremony with less make-up and a simple dress, unlike before, when she had to spend USD 1000 for a dress and a whole day dressing up, yet not confident and anxiously looking as if others were more beautiful than her. Interviewees now feel liberated because they are no longer enslaved by consumerism. *Aim* no longer feels anxious thinking about ice cream like a drug addict and he did not have to return home because he forgot to put on perfume.

## 5. General Discussion

Our findings describe the consumption behaviors and motivations of fifteen active Buddhist consumers in Bangkok, Thailand, who devote themselves to Buddhist practice to liberate themselves from suffering. Given that superordinate goal, their consumption mainly focuses on fulfilling their basic needs through sufficient consumption of products with functional value. Their consumption behaviors are reduced, simplified, purposeful, and less susceptible to external factors. However, Buddhist consumption does not equate to anti-consumption. Buddhists do not have to choose the cheapest consumption mode and forego joy in their lives. Sometimes, interviewees buy expensive goods if they satisfy utilitarian needs with high quality and durability. Unnecessary products are occasionally purchased to support others (e.g., the poor). Our findings reveal seven stages interviewees

go through before observing significant changes in their consumption. We identified mechanisms and strategies to progress through these stages, including cultivating emotional stability and deliberation.

To further explain how these mindfulness-based strategies help active Buddhist practitioners deal with desire and temptations, we drew on the dynamic model of desire from Hofmann and Van Dillen (2012). The model suggests automatic and deliberate processes by which desires influence behaviors (see Figure 2). The automatic process starts as the reward-processing system in the brain evaluates stimuli attractiveness against consumer motivation and learning history, resulting in impulsive behaviors (Kotabe and Hofmann 2015). The deliberate process starts as desire accesses the working memory. When consumers allocate more attention to a rewarding stimulus, they are more likely to experience a subjective feeling of craving or desire. As consumers deliberately reflect on their desire, they may construct justifications or excuses for indulging (deliberate processing) (Kavanagh et al. 2005). As a result, Hofmann and Van Dillen (2012) proposed that to prevent the development of desire, one must interfere with the early automatic processing of temptation before it enters the working memory, as it will become more difficult to regulate the desire.

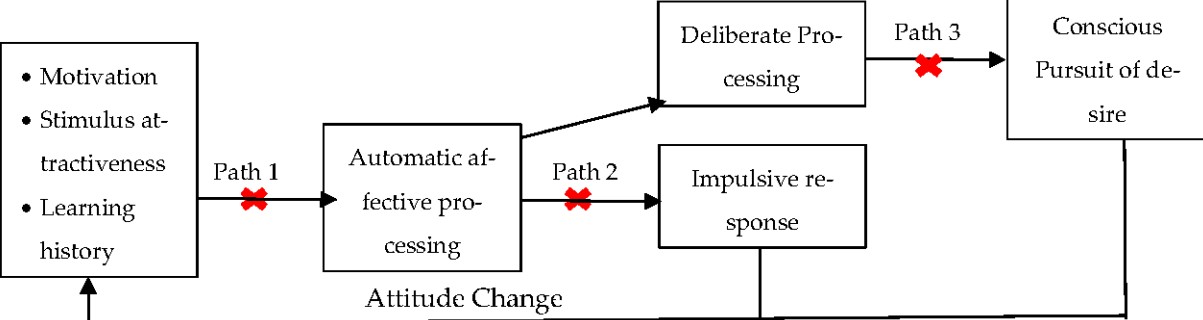

**Figure 2.** A Dynamic Model of Desire (adapted from Hofmann and Van Dillen 2012).

Active Buddhists manage to prevent desire in the early and subsequent steps as their lifetime goal and motivation have changed, resulting in a change in perception of stimuli characteristics and their learning and memory (*sañña*). Some interviewees in higher stages of practice are less emotionally sensitive when exposed to a pleasant stimulus. The affective reaction is attributed to their long-term practice, which alters their perception and feelings toward the stimuli and, therefore, automatic affective processing of rewarding stimuli seldom occurs (Path 1). The affective response may still occur for those in the lower stages of practice. However, they may be more conscious of their true physical and psychological states and not engage in mental fabrication. By observing the current emotional experience without reacting to it (equanimity), they may realize that the feelings and cravings for a rewarding stimulus can fade away and no longer react impulsively. Some interviewees reported that they could quickly let go of the impulse (Path 2). Very attractive stimuli may receive more attention and proceed to deliberate processing, where the costs–benefits, necessity, practicality, and specific functional criteria (price/quality) are evaluated further (Path 3). Therefore, active Buddhists can deal with temptations better with reduced emotional sensitivity and enhanced logical reasoning. Furthermore, a decline in automatic and conscious pursuit of desire gradually alters the interviewees' habits and attitudes towards desirable stimuli, resulting in substantial changes in their consumption behaviors.

This study provides several theoretical contributions. To begin with, we add to the small but growing number of mindful consumption studies (e.g., Bahl et al. 2016; Fischer et al. 2017; Gupta et al. 2021; Sheth et al. 2011; Milne et al. 2020). We extended the research of Milne et al. (2020), who studied three basic aspects of mindful consumption: economic aspect (mindful of price and quality), firm aspect (mindful of firm activities), macro impact (mindful of the impact of the entire process and systems on the society and environment)

by exploring consumption motives and behaviors at the individual level throughout the consumer journey and identifying patterns of consumption change to understand the role mindfulness plays in the consumer's life. As mindfulness practice has its roots in Buddhism, our paper extends existing research in Buddhist psychology, which provides insights into consumer behaviors but has been largely overlooked in the consumer research field. The extant research on mindfulness from the Buddhist or psychological perspectives provided a conceptual framework that is not sufficiently tied to consumer psychology and is not tested with empirical research (Bahl et al. 2016; Mathras et al. 2016; Mick 2017; Milne et al. 2020). Most empirical research on mindfulness also employs quantitative measures and mindfulness interventions that do not provide rich information to understand a highly nuanced and intricate topic such as mindfulness (Thiermann and Sheate 2021). This study responds to the need for empirical and qualitative research on mindfulness and mindful consumption (Thiermann and Sheate 2021). Specifically, it addresses the research agenda established in previous studies concerning how active Buddhist consumers consume and have gradually adjusted their consumption behavior (Thiermann and Sheate 2021), how mindfulness helps control temptation, their dependency on material goods and the self (Bahl et al. 2016; McCullough and Willoughby 2009), and how they can cherish consumption experiences without emotional attachment and ego defense (Mick 2017). Moreover, we elaborated on the mindfulness mechanisms proposed and discussed in Hölzel et al. (2011) and Bahl et al. (2016) and further delineated steps active Buddhist consumers go through to control their temptation. Interestingly, the last stage of behavioral change revealed that active Buddhists go beyond resisting the temptation to understanding the true nature of desires and temptation and being able to live in harmony between two worlds and break free from being directed by temptation.

As the COVID-19 pandemic and sustainability movement are paving the way for mindful and sustainable consumption (KPMG International 2020; Milne et al. 2020), marketers and policymakers need to understand the consumption behavior patterns of mindful consumers to cultivate mindful consumption. Lessons learned from active Buddhists who are mindful consumers provide the following managerial implications. First, mindful consumption is based on needs, necessity, and well-being. Marketing communication should be prevention focused (e.g., concerned with safety, security, and risk mitigation) rather than promotion focused (e.g., growth, advancement, and aspiration) (Higgins 1997). As mindful consumers tend to minimize time and effort in the purchase, they tend to pick the well-established brand with moderate quality and price that warrant durability and safety. The product information presented to them should focus on core functions and services, emphasizing reliability, durability, and maintainability. As they tend to express kindness and compassion towards others, they would support a company that cares for society, human welfare, and the environment. Brands should share information about the brand's meaningful purpose, minimal impact on others, and community involvement. As they tend to use reasons more than emotions and prefer consumption in the form of experiences rather than possessions, the functional and epistemic value would be preferred to hedonic and symbolic values of purchase. Companies should, therefore, invest more in meaningful activities, such as educational workshops, training, and community development, to enhance customer well-being and self-development and connect customers to society. Lastly, as mindful consumers have lower expectations and tend to repurchase from the same company, businesses can attract and sustain relationships with their customers by supporting their mindful consumption. Companies could, for instance, avoid inspiring consumers' ideal selves, recommend optimal consumption levels, commend customers who utilize products for better self-improvement rather than excessive spending, and provide service support for disposal, maintenance, repair, and reuse for other purposes.

This study is limited by the sample size and characteristics. Sample sizes of 20–30 are typically used to establish data saturation using a ground theory approach (Creswell 1998). Further, our interviewees tend to be highly educated and middle-class consumers in an urban city who are in sufficiently comfortable circumstances that enable them to live

a simpler life without much trouble. Further investigation of consumers from broader geographic and demographic backgrounds with larger sample sizes is encouraged.

**Author Contributions:** Writing—original draft, A.W.; Writing—review & editing, P.J. All authors have read and agreed to the published version of the manuscript.

**Funding:** This research received no external funding.

**Institutional Review Board Statement:** This study was approved by the Office of the Committee for Research Ethics (Social Sciences), Faculty of Social Sciences and Humanities, Mahidol University on 10 November, approval code: 2015 2015/352.1011.

**Informed Consent Statement:** Not applicable.

**Data Availability Statement:** Data sharing is not applicable to this article.

**Conflicts of Interest:** The authors declare no conflict of interest.

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
