# Peer review of "The Path of ‘No’ Resistance to Temptation: Lessons Learned from Active Buddhist Consumers in Thailand"

_religions, doi:10.3390/rel13080742_

Round 1

Reviewer 1 Report

This paper is on an interesting topic that would likely have appeal to readers of this journal. The literature on Buddhism and consumption is perhaps best outlined in the Conceptual Background section, however lacks sophistication and complexity especially in the introduction. I would like to see more engagement with more recent literature on Buddhist economics, in particular, demonstrating the authors' awareness of the ways Buddhism and economics and entwined in complex and contested ways, beyond a 'middle path' approach that treats Buddhist consumption as a matter of balancing between the two extremes of frugality and excess consumption, and which is present in Thailand as well as other contexts (e.g. see https://oxfordre.com/religion/view/10.1093/acrefore/9780199340378.001.0001/acrefore-9780199340378-e-694). The paper also requires an explanation of what is meant by an 'engaged Buddhist', likely with a description of how the authors' use of the term differs to standard definitions of engaged Buddhism. I question the use of the term in the paper, as it appears that participants may be more accurately be described as specifically middle class individuals influenced by Thai consumer culture, and coming from comfortable circumstances that enable them to curtail their spending from a previously high level. With participants ranging from 1 to 17 years of practice, the significant variation in length of practice, the study raises questions about how assessments about 'stages of behaviour change' (RQ2) are made about individuals practicing for different lengths of time. The conclusions and recommendations to marketers are interesting, however it is not clear to what extent 'mindful consumption' has risen in recent years, as the authors suggest (p. 22) - is there a source that can be referenced here?, and to what extent the prevalence (or not) of 'mindful consumers' warrants to attention of marketers. Overall I suggest that the article be revised for clarity of argument and English language accuracy. 

Reviewer 2 Report

Brief Summary:

An interesting investigation into active Buddhist practitioners’ experiences of disengaging from consumerist practices that draws on extensive interview materials for support. The authors propose a very compelling process for thinking about consumption through a Buddhist lens and engage with a variety of literatures to think through and support their proposed system. However, the authors do not engage with much of the recent literature that has been done about Buddhism and economics (including Buddhist economics) which seems essential for this work (e.g. Buddhism and Business, edited by Trine Brox and Elizabeth Williams-Oerberg and Monks, Morality, and Merit, edited by Christoph Brumann, Saskia Abrahms-Kavunenko, and Beata Świtek). The authors also do not employ “engaged Buddhism” correctly. The article seems to equate “engaged Buddhism” with active involvement in Buddhism. “Engaged Buddhism” is a very particular type of Buddhism with a plethora of literature associated with it (e.g. Engaged Buddhism: Buddhist Liberation Movements in Asia, edited by Christopher Queen and Sally King and “Introduction: Reformulating ‘Socially Engaged Buddhism’ as an Analytical Category” by Jessica Main and Rongdao Lai). The term needs to be eliminated in this paper unless the authors address engaged Buddhism itself. Along the same vein, the article continually uses “Buddhists” to refer to all Buddhists rather than contextualizing that this work happened in Thailand with a particular kind of Buddhist belief, practice, and history. The authors should be much clearer throughout about the context of the research and not imply over-generalizations. The conclusion is also too broad and veers into proselytization, that all people would benefit from this practice, i.e. becoming a very particular kind of Buddhist. It is apparent that this practice has been helpful for the interviewees and might be helpful for others but that doesn’t point to it being a universally applicable. The final paragraph has something interesting to say about how this information can be used in marketing but the explanation is not entirely clear, especially concerning its connections to the information presented.

Broad comments on strengths and weaknesses:

Strengths

·         Compelling idea for the paper and excellent interview materials interwoven throughout the data section.

·         The literature review (section 2) is very well written and brings together a lot of varying information from disparate fields into conversation with each other.

·         Set-up for the project and data collection methods are sound for what the authors’ wish to research.

Weaknesses

·         The entirety of Section 2.1 is unnecessary. The paper engages with these ideas in a much more useful way in lines 167-196 in Section 2.2. Only a few modifications to that section will provide the reader with the necessary background for understanding basic Buddhist positions related to materialism and dukkha.

·         Throughout, the word “Buddhism” is used as an adjective when it is not. “Buddhist” is the adjectival form of the word and all instances should be checked. I mentioned a few below as examples but this needs to be fixed. (e.g. Buddhist values, Buddhist practice)

·         The interview excerpts from section 4 are excellent but need to be analyzed in much more detail. They’re just put out there for the reader to think through but even the interviewee does not always mention that Buddhism is the reason for their changes. The authors needs to make those connections clear to the reader. It reads as though the authors came up with an idea and found quotes to support it other than grounding their analysis in the quote provided.

Specific comments with line numbers, tables, or figures:

Line 21, 47, 51, 57: “Buddhist practice” not “Buddhism practice”

Line 28: “Make merit” not “Make a merit”

Lines 62-65: This paragraph is perfunctory. It should either do more or be deleted.

Line 72: “what we crave.” “for” is not necessary.

Lines 82-85: Are the lists of cravings in the 2nd Noble Truth quotations or translations by DeSilva? If not, the citation seems unnecessary.

Line 85: Are you linking the forms of craving to these consumptions? That seems like a larger project than just stating it. Explain those connections more.

Line 89-90 and 98: For Noble Truths 1 and 2, the authors mention the Pali names for the stages but not for 3 and 4. Be consistent and name all or none.

Lines 98-113: The discussion of the 8-fold path in totality seems excessive. Focus on what’s important for your argument here and include that only.

Line 125: The use of “you” is jarring.

Line 127-132: This paragraph seems to imply that Buddhism itself is not consumeristic which conflicts with much of the recent work in the area, especially Buddhism and Business (edited by Trine Brox and Elizabeth Williams-Oerberg) which the authors need to engage with.

Line 198: “consumers” not “csumers”

Line 198-212: It is unclear if this paragraph is about all consumers or only Buddhist ones.

Line 255: “has” not “have”

Line 290: “leads” or “led” not “lead”

Line 334: How does one judge that a Buddhist “regularly and genuinely practices” Buddhism? That seems like a value judgement (especially “genuinely”) that the authors do not defend.

Line 339: “Theravada branch” not “the Theravada Buddhism”

Line 342: Liusuwan is a strange citation here. The cited article does not provide the information the authors claim it does and regardless, it is not peer-reviewed or even factual but rather a think piece.

Line 343: What is the “King’s Sufficiency Economy” and how does that influence what’s going on in this study?

Lines 370-382: This makes it sound like there will be ethnography involved in this paper which there isn’t. In addition, it’s not necessary to the paper and clouds the methodology. Delete for clarity.

Line 500-501: I’m not sure what the authors mean by “become malfunctioned.” Maybe “easily break”?

Line 532: “go out and shop” not “shopping”

Line 558: “truth of life” here feels value-laden. “Buddhist truth” or something of the sort would be more appropriate.

Lines 677-680: The transformation of the meditation teacher seems like it needs to be explored much more in-depth. The transformation here is an emotional as well as consumptive/economic one that would benefit from more explanation.

Line 683: “addicted” has a negative connotation which doesn’t seem to fit the rest of the sentence and paragraph.

Line 704: I think “that” is unnecessary in this sentence, otherwise it’s a fragment.

Line 731-732: The meaning of this sentence is unclear.

Line 816-819: Mentioning impermanence in “Outer Happiness did not last long” seems essential.

Line 942: What is a “future research request”?

Reviewer 3 Report

The authors are passionate about their hypothesis that those who do a serious Buddhist practice become much less consumer-oriented.  According to this hypothesis, these practitioners become less influenced by consumer trends, and more likely to resist such activities as drinking in bars, listening to emotion-laden music, and buying fashionable clothes. 

The authors’ methodology for validating this hypothesis is to interview a group of people, many of them the authors’ friends or acquaintances, about how their personal attitudes and behaviors had changed after doing some type of Buddhist practice.  Interviewees had practiced for different lengths of time and under different teachers.  The personal statements of those interviewed were presented as validation of the authors’ hypothesis that Buddhist practice results in changed consumer behavior.

Although the authors explain that their research is qualitative, their conclusions would be stronger if they had recruited interviewees with whom they had no prior relationships.  The reader wonders if the authors may have chosen interviewees who already agreed with their core hypothesis, or whether the authors unconsciously influenced the responses of the interviewees in the context of their friendship.  To strengthen and broaden their analysis, the authors might also address two related issues:  other possible reasons for the noted behavior changes; and the length of time the self-avowed changes have persisted.

Finally, the paper needs editing for English sentence structure and grammar.  In some places, problems with English grammar and structure result in confusing or contradictory statements. 

·       P. 3, line 122-23:  . . . when happiness is deprived, people become suffering.

·       P. 2, line 76-77:  Wants cannot be satisfied if it is not fulfilled, and we cannot fulfill every want.   [unclear what “it” references]

·       P. 17, line 750-51:  . . .  emotional regulation can lead to a failure of self-control.

When citing other sources, research should be cited in a consistent way.  For example, the authors refer to the writer Thanissaro Bhikkhu sometimes with his full name and other times as if Bhikkhu were his surname rather than his title as a monk.  Also, the article would be strengthened by citing references only to more recent scholarship, i.e. last 20 years or so, unless there are specific reasons for using older sources. 

The term “engaged Buddhist” should be defined in the beginning of the article, as many cultures use this term to indicate Buddhists engaged in social activism, a meaning not intended by the authors. 

Overall, this article is quite interesting, in particular as it presents the reflections of individual Buddhists on how Buddhist practice has changed their perspectives on consumerism.  Readers may find the insights of Thai Buddhist practitioners quite helpful as they study the relationship between Buddhist teachings and specific behaviors.  Indeed, the authors might consider reframing this article to highlight these observations and reflections, rather than structuring the article as qualitative research with hypothesis and conclusion.  

Reviewer 4 Report

This manuscript is a qualitative approach concerned with showing how Buddhist practice and teachings help curb sensual/ material temptations and thus reduce unnecessary consumption. This research is meant to show the stages one goes through when changing behaviour regarding consumption and the mechanisms which are utilized to progress through these stages. Overall, I agree with this project, and I enjoyed reading it. This is a very important topic, especially in the climate of global capitalism where consumerism has become a way of life. As this paper wisely states, Buddhism has a lot to offer the modern world. Overconsumption is a source of suffering that, with practice can be tamed.

            That said, below are a series of concerns that arose while reading this manuscript.

Primary Concerns:

This paper is over 15,000 words in length. It has a lot of promise however, the valuable content is dampened by a lot of transcription. Paraphrased interviews in the findings section would be helpful. Rather than reading what practitioners said it would be more fruitful to know what the author(s) have interpreted from the interviews. An ideal length for a paper like this would be 8,000-10,000 words.

It is not clear how or what the practitioners actually did. What specific Buddhist teachings and practices were considered? Did they vary among the participants? The emphasis of this paper is on the subjective account of the practitioners after having engaged with Buddhism however, “engaged with Buddhism” seems to mean maintaining regular practices by reading books, meditating, listening to Dhamma talks, and attending retreats every year (lines 357-358). These forms of engagement are quite general. This paper would strongly benefit from a more detailed breakdown of the content of these books, Dhamma talks, and retreats. Although lines 608-610 state that the participants have to experiment and find their own suitable methods of practice it would be helpful to have an outline of what the practices could consist of.

Details regarding the inductive approach (line 370) the author(s) took are important. While the author(s) have included details of various interviews details of the analysis itself would be indispensable. It is said that recurring themes, concepts, patterns, assigned codes, etc. were considered however the details of the method are not explained. What are the themes and patterns that were found? How were they found? Why are these themes and patterns significant?

In lines 940-941 the author(s) writes that this paper aims to provide a comprehensive framework and empirical analysis to bridge the gap between Buddhist psychology and consumer psychology. While the framework exists as a seven-stage model it is unclear how this model is superior to similar models that came before it.

It is not until page 12 that it becomes clear what the core of this research project is. Lines 528-529 state that the stages of behaviour change found by analyzing the interviews are distinct from previous models: Transtheoretical Model (TTM) and Stepwise Model of Behavioural Change (SMBC). Although this paper plans to show how Buddhist practice reduces temptation and consumption, it seems that this research is also concerned with a Buddhist interpretation of TTM and SMBC models. This Buddhist interpretation is a crucial part of the project and therefore should be included in the abstract and introduction.

The author(s) assume the perspective of Buddhist normativity using terms like “facts” and “truth” referring to various teachings. Despite the English translation of cattāri ariya-saccāni as “the four truths” absolute statements using terms like “fact”, “truth”, “right and wrong”, and “reality” should be reconsidered when referring to philosophical interpretations (see lines 68, 110, 112, and 169 respectively). Furthermore, on lines 850-851 it is stated that “[a]fter practice, they learned to think, speak, act positively, which make them become a better person.” One must assume that “better person” here refers to one who consumes in a way that aligns with Buddhist precepts. The author(s) need to explicitly state that they are operating within the spectrum of Buddhist normativity. Finally, a similar concern comes on lines 776-777 where it is stated that “interviewees become less sensitive to marketing stimuli….” This kind of statement cannot be made without sufficient evidence. In the current context it would have to be stated “interviewees reported feeling less sensitive to marketing stimuli…”

Secondary Concerns:

The abstract would benefit from being longer (about 200 words). It could mention that a major focus of the research is the development of a Buddhist-based interpretation of models for changes in behaviour (ie. TTM and SMBC).

It would be helpful to define the term “engaged Buddhism” early in the essay. “Engaged Buddhism” generally refers to socially or politically engaged Buddhism whereas it is being used in this manuscript to denote one who actively practices Buddhism.

This manuscript is written in English that is relatively easy to understand however, it requires proofreading for grammatical mistakes. For example, on page 1 the articles “a” and “the” are incorrectly used (lines 17, 29, 43…) Also “Buddhism practice” (lines 6, 21, 47…) needs to be “Buddhist practice”.

Although many publications do not include them, diacritics should be included in an academic paper. For example, tanha --> taṇhā (line 81)

On page 6 section 2.4 is missing.

Subheadings in section 4 need to be corrected. For example, 2.1 --> 4.1 (line 394)

Stage five “tight and tear” is a bit unclear and could use some explanation as to what that term means.

The list of references requires hanging indentation.

Suggestions:

Less emphasis on the content of the interviews and more explanation by the author(s). In order to reduce the length of the manuscript and to clarify the interviews it would be good to remove or paraphrase some of the transcribed content.

If the author(s) want to theorize about how Buddhist practices can support the reduction of unnecessary consumption through behaviour changing practices they can do so without subjective accounts of interviewees. However, if the author(s) want to show evidence that Buddhist practices can reduce consumption through behaviour changing practices then they must include a more detailed account regarding the practices themselves (what kind of meditation was being utilized?), as well as more detail regarding the analysis of the data.

Round 2

Reviewer 2 Report

The paper is much improved. It is clearer, it focuses on more the contribution to scholarship and adds in some much-needed engagement with Buddhist economics in the introduction. There are still a few issues sprinkled throughout:

Lines 37 & 40: These sentences both start with “However” which is confusing.
Line 543-544: I don’t think Thailand is considered the “World’s leading shopping destination.”
Line 589: “All interviews were” not “all interview was”
Line 735: The use of Brox and Williams-Oerberg here is incorrect. The argue that there is that perception but it is not correct.
Line 1588 and 1562-1563: These suggestions seem to imply that the audience of this article is marketers which is not the case for this journal.

Reviewer 3 Report

The authors have done a remarkable job, responding to the comments of the reviewer and submitting a substantial reframing of their research.  This newly edited analysis and discussion offers the reader a much clearer presentation of the authors’ hypothesis, evidence, and reasons for conclusions.

This paper can be strengthened in the following ways:

·      One more edit for clearer English grammatical structures

·       Make sure the “tables” and “figures” are labelled correctly, that the numbers of the tables/figures match the references to them in the text.  [e.g. See p. 31 reference to Figure 1.  Not clear where to find Figure 1.  Is this a reference to Table 1 on p. 8? ]

·       Discussion of the two Figures on p. 32 might be a bit clearer in explaining the relationship of thoughts and feelings to desire and consumer behavior

I look forward to seeing more work from these researchers.

Reviewer 4 Report

The 16 comments given in the first round were all taken into consideration by the authors and as a result the paper is much improved. The most significant changes are mentioned below.

The paper length was reduced and in so doing the aim of the paper became clearer. While the content of the paper was reduced there has also been significant additions. The inclusion of Buddhist economics is particularly well suited to the content of this paper. Also, the reduction in transcription and increase in interpretation is crucial. Moreover, the practices of active Buddhists were appropriately defined which gives the reader the necessary clarity regarding the interviewees' practices.

Lines 841-915 (stage 2) give an account of the type of practices that were being done including concentration and insight meditation. This is important. The adjustments and additions made in stage 4 are a further support to the details of the practices.

In response to comment 16, the authors point out that they were hesitant to make the paper “too religious”. Not all epistemically relevant content comes from quantitative data. The journal in question, after all, is entitled Religions. The merit of this manuscript comes in the form of the interpretation of interviews and expressions of interviewees based on their lived experiences. I applaud the decision of the authors to rework the methodology section to include a more unapologetically religious and qualitative approach.

Finally, including limitations of the work is important. This adds to the overall merit of the paper and helps to guide future research in this area.

Minor corrections:

There is an assortment of minor corrections which include but are not limited to the following.

Line 14 – “urban city” “urban areas” or “city”

Line 270 – start starts

Line 370 – consumption Consumption

Line 378 – Table 1 summarizes the four foundations unfold Table 1 summarizes how the four foundations unfold

Line 419 – There seems to be an unnecessary space dividing these paragraphs

Line 451 – include but not limited to meditation practice… includes but is not limited to meditation practice…

Line 550 – Buddhism practice Buddhist practice

Lines 676, 714, 751 – consider removing “?”

Line 870 – check parenthesis

Line 1046 – “how difficult he experienced when he tried to observe…” “how difficult it was when he tried to observe…”

Line 1179 – “observe the temptation to disappear” “observe the temptation until it disappears”

Line 1425 – reconsider spacing

Line 1466 – they can quickly let of the impulse they can quickly let go of the impulse

Line 1538 – COVID-19 Covid-19 (to match use of term on line 32)
